# High-Risk Genotypes of Human Papillomavirus at Diverse Anogenital Sites among Chinese Women: Infection Features and Potential Correlation with Cervical Intraepithelial Neoplasia

**DOI:** 10.3390/cancers16112107

**Published:** 2024-05-31

**Authors:** Chao Zhao, Jiahui An, Mingzhu Li, Jingran Li, Yun Zhao, Jianliu Wang, Heidi Qunhui Xie, Lihui Wei

**Affiliations:** 1Department of Obstetrics and Gynecology, Peking University People’s Hospital, Beijing 100044, China; 2State Key Laboratory of Environmental Chemistry and Ecotoxicology, Research Center for Eco-Environmental Sciences, Chinese Academy of Sciences, Beijing 100085, China; 3University of Chinese Academy of Sciences, Beijing 100049, China

**Keywords:** high-risk human papillomavirus (HR-HPV), anogenital, prevalence, cervical cancer

## Abstract

**Simple Summary:**

Cervical cancer is one of the most prevalent cancers among women. However, the clinically well-accepted cervical HPV 16/18 test was not sufficient to screen patients with high-grade cervical intraepithelial neoplasia (CIN 2+). The purpose of this study was to investigate the infection features of 15 high-risk (HR) HPVs in cervix, vagina, vulva, and anus, and evaluate their potential association with cervical lesions. In this study, HPV genotyping was performed on samples from the four anogenital sites of 499 Chinese women. Results showed that in addition to the well-recognized cervical-cancer-associated subtypes, e.g., HPV 16, 52, and 58, HPV 51, 53, and 56 had high prevalence and site-related infection consistency among the anogenital sites studied. In addition to cervical HPV 16/18, cervical HPV 33/35/52/53/56/58 were suggested as potential indicators that could improve the accuracy of HR-HPV screening in predicting CIN 2+.

**Abstract:**

Background: Both cervical cancer and cervical intraepithelial neoplasia (CIN) are associated with human papillomavirus (HPV) infection at different anogenital sites, but the infection features of high-risk (HR) HPVs at these sites and their association with cervical lesions have not been well characterized. Given the limitation of cervical HPV 16/18 test in screening patients with high-grade CIN (CIN 2+), studies on whether non-16/18 HR-HPV subtype(s) have potential as additional indicator(s) to improve CIN 2+ screening are needed. Methods: The infection of 15 HR-HPVs in vulva, anus, vagina, and cervix of 499 Chinese women was analyzed, and CIN lesion-associated HR-HPV subtypes were revealed. Results: In addition to the well-known cervical-cancer-associated HPV 16, 52, and 58, HPV 51, 53, and 56 were also identified as high-frequency detected subtypes prevalently and consistently present at the anogenital sites studied, preferentially in multi-infection patterns. HPV 16, 52, 58, 56, and 53 were the top five prevalent subtypes in patients with CIN 2+. In addition, we found that cervical HPV 33/35/52/53/56/58 co-testing with HPV 16/18 might improve CIN 2+ screening performance. Conclusion: This study provided a new insight into HR-HPV screening strategy based on different subtype combinations, which might be used in risk stratification clinically.

## 1. Introduction

Cervical cancer is the fourth most common female malignant tumor globally, which seriously threatens female health [1]. More than 95% of cervical cancer cases are attributed to persistent HPV infection [2]. To date, 15 high-risk (HR) HPV subtypes have been identified [3], among which infections to HPV 16, 18, 31, 33, 35, 39, 45, 51, 52, 56, 58, 59, 68, 73, and 82 contribute to 96.6% of invasive cervical cancers [4]. In addition to the cervix, other areas of the female genital tract (e.g., the vagina, vulva, and perineum) and anogenital mucosae could also be infected by 40 HPV subtypes [5]. The epithelial transition zones, such as the endo-/ectocervix and anorectal junctions, are regions more susceptible to carcinogenesis by HR-HPV types [6]. Studies also revealed that being HPV positive at the cervix extensively increased the chances of being positive at other sites [7]. Thus, a better comprehension of the infection characteristics at other anatomic sites beyond the cervix may help us explain the natural history of HR-HPV infection as well as improve the strategy of HPV screening in cervical lesions.

Up to now, data on the prevalence and distribution of HR-HPV infection at different anatomical sites outside the cervix (e.g., vagina, vulva, anus, and/oral mucosa) have been reported in women from Germany [7], Rwanda [8], Australia [9], the United States [10], and Croatia [11], as well as globally [12,13,14]. Besides the distribution of HR-HPV subtypes at different anogenital sites, these studies also provided information about the contribution of anatomical HPV infection to corresponding genital tract tumors, such as vaginal cancer [8,10,12], vulvar cancer [8,10,12,13,15], and anal cancer [8,12,14]. Some studies also evaluated the concordance of HR-HPV detection in matched urine, vaginal, and cervical samples in women from Korea [16], England [17], Belgium [18], and Denmark [19], which suggested that urinary HR-HPV detection offered an alternative strategy of cervical HR-HPV screening [17]. It has been shown that HR-HPV screening at anogenital sites other than the cervix can also help improve the sensitivity and efficiency of early screening in patients with cervical high-grade squamous intraepithelial lesions (HSIL) [20], particularly for women wanting to avoid or having difficulty in cervical biopsies for HR-HPV testing [21]. However, there is still a lack of studies on characterizing HR-HPV infections at different anogenital sites and their correlation with cervical lesions.

The routinely used HPV screening, which only indicates positive infection status, does not necessarily predict pathological changes. Thus, with the change in cervical cancer screening strategies, HR-HPV-based risk assessment and management are going to be gradually incorporated into routine cervical cancer screening [22]. In this context, studies on the correlation between the HR-HPV infection(s) and cytology/pathology features of individuals have been carried out. It has been shown that women infected with multiple types of HR-HPV are more likely to have abnormal cytology [23], and even suffer from a subsequent increased risk of cervical cancer or precancers [24]. Among the HR-HPVs, although HPV 16 and 18 are the most commonly detected subtypes of cervical cancer and precancerous lesions in women, HR-HPV infection other than HPV 16/18 contributes 25%–35% to cervical cancer [25]. According to the 2020 Updated Guidelines For Management Of Cervical Cancer Screening Abnormalities issued by the American Society of Colposcopy and Cervical Pathology (ASCCP) [26], colposcopy is used to manage women with HPV 16/18 infection(s), even if with negative cytology; for non-HPV 16/18 infected patients with negative cytology, a re-testing for a combination of HR-HPV (including HPV 16/18 and non-16/18 HR-HPV) and cytology is recommended one year later. Population-based prevention strategies should give preference to HPV 16/18 [27]; however, there is no consensus regarding the best risk stratification strategy for non-16/18 HR-HPVs-infected women [28]. It is suggested that non-16/18 HR-HPV infections, especially HPV 52 and 58, are also frequently detected in Chinese women, particularly those with HSIL [29]. Thus, primary HPV screening with HPV 16/18 genotyping has proved feasible but a new stratification strategy should be developed to increase the accuracy for predicting patients with HSIL. However, the reason for the lack of consensus on a risk stratification strategy could be complex. On one hand, there are insufficient data on the single/multiple-infection status of HR-HPVs, especially for infections other than HPV16/18 [12]; on the other hand, although some findings have suggested that the association between multiple HR-HPV infections and HSIL and low grade squamous intraepithelial lesions (LSIL) were stronger compared to single HPV infections [30], until now there has been no uniform conclusion on which specific combinations simultaneously affect the risk of HSIL and LSIL. Thus, study on HR-HPV genotyping and the lesion-related HR-HPV clusters is necessary to meet the requirements of predicting risks and to guide treatment and follow-up decisions for women with atypical squamous cell of undetermined significance (ASC-US) and LSIL [22].

Cervical cancer is the second most common gynecological malignancy in China [31]. It is estimated that 130,000 cases are newly diagnosed annually in China, accounting for approximately 28% of the total number of new cases of cervical cancer worldwide [31]. Although there have been systematic reviews of the characteristics of HR-HPV infection in Chinese women, few studies have focused on the distribution of HPV genotypes outside the cervix. Apart from limited studies on the vagina [32] and vulva [33], little was known about the anal HPV distribution in Chinese women and its relationship to cervical lesions. Evidence showed that the distribution of vulvar and cervical HPV was similar in patients with CIN 2+/CIN 3+, which indicated that vulva-based HPV showed comparable sensitivity and specificity to cervix-based HPV in the detection of CIN 3+ [33]. Moreover, Zhang et al. provided evidence for the consistency of HR-HPV subtypes and viral loads among the lower vagina, upper vagina, and cervix as well as the correlation between HR-HPV infection and clinical outcomes [34]. Based on that, Li et al. further included samples from the perineum site and found that the agreement of the HR-HPV infections in cervix and perineum was 79.35% [35]. However, the HR-HPV infections at different anogenital sites and their single/multiple-infection status have not been well characterized in Chinese women. Due to the significant regional differences in the distribution and prevalence of HR-HPV subtypes [36], the data from other countries can hardly be used as a reference for Chinese women. Therefore, the purpose of this study was to reveal the characteristics of single/multiple HR-HPV infections in Chinese women and the multi-infection pattern of HR-HPVs at four anogenital sites, including vulva, anus, vagina, and cervix, and to analyze the performance of different screening strategies regarding both HR-HPVs and anogenital sites combinations, so as to provide new clues for exploring adequate HR-HPV tests for HSIL screening.

## 2. Materials and Methods

### 2.1. Study Population

This study was performed at the Department of Gynecology and Obstetrics of the Peking University People’s Hospital (from June 2022 to September 2022). Inclusion criteria for participants were 25~64-year-old women with ≥ASC-US and/or HPV infection according to ThinPrep liquid-based cytological test (TCT) and HPV screening reports, who voluntarily participated. Exclusion criteria: women during the acute phase of any disease, pregnant, diagnosed with cervical lesions or cervical cancer, with previous hysterectomy and cervical resection, with acute or recurrent genital and urinary tract infections, or with pelvic radiation therapy. Eventually, 499 women were recalled for sample collection at four anogenital sites and colposcopy examination according to inclusion criteria (mean age 41.1; range, 19–67) (Figure 1). All patients signed the written informed consent. This study was approved by the Ethics Committee of Peking University People’s Hospital (ethical approval number: 2022PHB045-001).

### 2.2. Cervical Cytological Test, HPV Screening, and Colposcopy

Cervical exfoliated cells were collected to conduct TCT and HPV screening as reported previously [37]. Briefly, a vaginal speculum was placed to expose the cervix, and cervical exfoliation was performed at the squamocolumnar junction of the cervix using sampling brushes for TCT and HPV screening separately. The sampling brushes were then subjected to a 20 mL PreservCyt1 solution (Hologic, Marlborough, MA, USA, DOC sample) for TCT, or placed into a 2 mL liquid-based medium (Jianyou Medical Tech Co., Ltd., Danyang, China) for HPV screening. Cytology results were reported according to the Bethesda 2014 classification system: negative for intraepithelial lesion or malignancy (NILM), ASC-US, atypical squamous cells cannot exclude HSIL (ASC-H), LSIL, HSIL, squamous cell carcinoma (SCC), adenocarcinoma in situ (AIS), and adenocarcinoma (ADC). Type-specific HR-HPV viral genotyping was detected through a polymerase chain reaction (PCR)-based multiplex HPV DNA genotyping kit (Jianyou Medical Tech Co., Ltd., Danyang, China). Women positive for any HPV or with cytological ASC-US+ were recalled for colposcopy examination and biopsy if any abnormality was observed. Pathology results were reported as normal, cervical intraepithelial neoplasia grade 1 (CIN 1), grade 2 (CIN 2), grade 3 (CIN 3), microinvasive carcinoma (MIC), SCC, AIS, and ADC [38,39].

### 2.3. Sample Collection at Four Anogenital Sites

Exfoliated cell samples at four anogenital sites were collected during colposcopy examination for the 499 participants mentioned above, who were instructed to avoid intravaginal medication or douching, and sexual intercourse for 72 h prior to the examination. All samples were collected during non-menstrual period. The patient lay in a lithotomy position, with vulva and perianal area fully exposed. Exfoliated cells at each anogenital site were collected separately with a cotton swab (Gaotai, Heze, China) or cytobrush (Jianyou Medical Tech Co., Ltd., Danyang, China). The specific cotton swab was to be rotated at vulva site (including medial sides of labia majora, specific labia minora, and posterior labial commissure) 3~5 times for vulva sample collection, or used for collecting anus samples 3~5 times at the surface of anus and perianal region, which were then preserved in cell preservation solution (Jianyou Medical Tech Co., Ltd., Danyang, China) for HR-HPV genotyping later. The specific cytobrush was used to collect the exfoliated cells from vaginal site (including upper vagina, lower vagina, anterior fornix, and posterior vaginal fornix) or cervix site, which were then preserved in the same cell preservation solution as previously mentioned. Ultimately, HR-HPV genotyping was performed on samples collected from the vulva, anus, vagina, and cervix sites of 499 women.

### 2.4. HR-HPV DNA Genotyping 

The cell preservation solution was subjected to HR-HPV DNA genotyping using a type-specific PCR kit (Sansure, Changsha, China) by an iPonatic S-Q31B system (Sansure, Changsha, China) according to the manufacturer’s instructions. The genotyping system includes detection of 15 HR-HPVs (including HPV 16, 18, 31, 33, 35, 39, 45, 51, 52, 53, 56, 58, 59, 66, and 68) and has been approved by the European Union certificate [40]. Briefly, the sample was centrifuged and 50 μL supernatants were reserved. Then, the 50 μL sample/negative control/positive control was mixed with 50 μL nucleic acid-releasing agent before being added into the PCR mixes. PCR was run by the iPonatic S-Q31B system. HPV positivity is measured by the cycle numbers observed (Ct) when the fluorescent signal reaches the set threshold. A Ct ≤ 39 is considered high-risk HPV positive and a Ct > 39 is considered high-risk HPV negative.

### 2.5. Statistics

The data are presented as the number of cases or percentage of participants. Cohen’s kappa statistic is a measure of agreement between two categorical variables [41]. Here, it was performed per event, in which an event was considered to be a specific analyzed subtype call or a negative call between two different anogenital sites [42]. Based on that, the agreement of occurring by chance between different anatomical sites was determined by Cohen’s kappa coefficient (κ) with 0 = poor, 0–0.20 = slight, 0.21–0.40 = fair, 0.41–0.60 = moderate, 0.61–0.80 = substantial, and 0.81–1.00 = (almost) perfect agreement [43]. Fisher’s exact test was used to compare categorical variables, including differences in the rate of occurrence of 15 HR-HPVs in single and multiple infections at each anogenital site. Moreover, the receiver operating characteristic (ROC) curve analysis could compare sensitivity versus specificity of selected index tests in order to evaluate their clinical diagnostic or predictive performance [44]. Sensitivity means the proportion of subjects with target condition (reference standard positive) and giving positive test results while specificity is the proportion of subjects without the target condition and giving negative test results [45]. Here, the ROC curve analysis was used to compare the CIN 2+ predictive performance of different HR-HPV screening tests by area under curve (AUC) value. The value for AUC ranges from 0 to 1. A model with an AUC of 1 is able to perfectly classify observations into classes while a model with an AUC of 0.5 does no better than a model that performs random guessing [46]. A *p*-value < 0.05 was considered as significant. Analyses above were performed using SPSS (version 27.0) and Origin (version 2023). Here, in order to analyze and visualize different combination models of HR-HPV at different anogenital sites or different cervical disease categories, R (Version 4.3.1) and an open-source R package, UpSetR, were used. UpSetR, developed by Conway et al. [47], employs a scalable matrix-based visualization to show intersections of sets, their size, and other properties [47], and is available at https://github.com/hms-dbmi/UpSetR/ (accessed on 20 December 2023).

## 3. Results

### 3.1. Concordance of the Infection of 15 HR-HPVs at Different Anatomical Sites

Of all 499 patients tested, 90 (18.04%) had no HR-HPV infection at all four sites, and 409 (81.97%) had at least one subtype of HR-HPV infection, among which 61 (12.22%) had HR-HPV infection at one site, 54 (10.82%) at two sites, 139 (27.86%) at three sites, and 155 (31.06%) at all sites. HR-HPV prevalence in cervix (74.15%) was the highest, followed by vagina (72.55%), vulva (59.72%), and anus (39.48%) (Table 1).

The distribution of 15 HR-HPVs at different sites is shown in Table 2. The top six HPV infection genotypes were HPV 16, 52, 51, 53, 58, and 56 in vulva. HPV 52 was the most common subtype in the anus, followed by HPV 16, 51, 53, 56, 58, and 39 (HPV 58 and HPV 39 tied for sixth place). For vagina and cervix, HPV 16 was also the most common subtype, followed by HPV 52, 58, 53, 51, and 56. Since HPV 18 has been reported as one of the most commonly found HR-HPVs in cervical cancer [48], we selected HPV 16, 51, 52, 53, 56, and 58, the most commonly detected subtypes of this study, together with HPV 18, to analyze the agreement of HR-HPV subtype distribution between any two anatomical sites by determination of κ (Figure 2). Basically, the agreement between vagina and vulva as well as the agreement between vagina and cervix were higher than the agreement between vulva and cervix for these six HPV subtypes, suggesting that the HR-HPV distribution at vagina might reflect that at vulva or at cervix. There was at least a substantial level of agreement (κ > 0.60) between vulva, vagina, and cervix regarding HPV 16, 51, 52, 53, 56, and 58. However, there was just moderate agreement (0.46 < κ < 0.62) for HPV 18 between vulva, vagina, and cervix, and slight or even fair agreement (0.12 < κ < 0.28) for HPV 18 between anus and the other three sites.

### 3.2. Single- and Multiple-Infection Features of 15 HR-HPVs at Four Anatomical Sites

The feature of single and multiple infections at four anatomical sites is presented in Figure 3. The single HR-HPV infection rates in cervix, vagina, anus, and vulva were 63.5%, 61.1%, 47.7%, and 46.3%, respectively. The maximum HR-HPV subtype number of the co-infection in vulva and anus was five, while a six-HPV co-infection of HPV 16, 18, 35, 53, 56, and 58 was observed in one patient’s cervix and vagina (Appendix A). Additionally, the three-HPV co-infection occurrence rate in vulva (17.1%) was significantly higher than that in vagina (9.7%) and cervix (7.5%).

Among 15 HR-HPVs, HPV 16 occurrence in single infection was significantly more frequent than that in multiple infections, regardless of anatomical site (Figure 4). The anal, vaginal, and cervical infections of HPV 18 occurred more in single infections than in multiple infections, while HPV 18 occurred more in multiple infections in vulva. The other 13 subtypes all preferred to co-infection rather than single infection at all four sites, although only significant differences were observed in HPV 35, 39, and 51 in vulva, HPV 33 in anus, HPV 51, 53, and 66 in vagina, and HPV 35 and 51 in cervix (Appendix A). 

It was noticeable that vaginal and cervical infections had similar HR-HPV infection features, including similarities in single/multiple-infection allocation (Figure 3) and subtype distribution (Figure 4). The dual infection clusters of the HR-HPVs were analyzed and visualized by Verver (R package), as shown in Figure 5. The most common single infection was HPV 16 in both vagina and cervix. The most common dual infection clusters were HPV (16 + 53) in vagina and HPV (52 + 56) in cervix. HPV (52 + 58), HPV (16 + 51), and HPV (51 + 52) dual infections were also relatively common. Additionally, the triple infection of HPV (52 + 53 + 56) occurred both in vagina and cervix. Taken together, compared to multiple infections, the single-infection patterns at these two sites were more consistent.

### 3.3. The Occurrence Feature of the 15 HR-HPVs in Different Cervical Lesion Categories

Based on the results of colposcopy and pathology tests for the 499 subjects, we found that 51.3% (256/499) were normal and the overall prevalence of precancerous lesions was 48.7% (243/499), of which CIN 1 was 37.1% (185/499), CIN 2 was 8.4% (42/499), and CIN 3 was 3.2% (16/499). The cervical HR-HPV infection rate was 23.0%, 74.05%, 97.6%, and 100% for normal, CIN 1, CIN 2, and CIN 3, respectively; while the overall HR-HPV infection rate at all testing sites (at least one subtype positive at any site) was 78.1%, 82.7%, 97.6%, and 100% for normal, CIN 1, CIN 2, and CIN 3, respectively.

To reveal the contribution of 15 HR-HPVs to cervical lesions, the total number of positive results for the 15 HR-HPVs at four testing sites was calculated for subjects with different cervical lesions (normal, CIN 1, and CIN 2+) (Appendix A). In subjects with normal or CIN 1, the predominant HR-HPVs infections at the four sites were HPV 16, 52, 53, 51, 58, and 56, which ranked differently, while HPV 51 was absent in the top six predominant HR-HPV subtypes in CIN 2+ group, in which HPV 16, 52, 58, 56, 53, 33, and 35 ranked differently. In addition to HPV 16, 33, 35, 52, and 58, which were already verified as having a strong relationship with cervical cancers [49], we also observed HPV 53 and 56 as relatively common. 

Interestingly, the average ratios of the total number of HPV infections regarding four anogenital sites to the number of patients with one specific HR-HPV subtype infection were almost more than 2 for these common subtypes, especially for those CIN 2+ patients (Table 3). Previous results also provided evidence that cervix shared at least a substantial level of agreement (κ > 0.60) with vulva and vagina regarding HPV 16, 52, 53, 56, and 58 (Figure 2). Here, we assumed that these prevalent HPV infection subtypes would occur at multiple anogenital sites for most CIN 2+ patients. It is clear that for any of the HR-HPV subtypes 16, 18, 33, 35, 52, 53, 56, and 58, the majority of infections occurred at cervix, vagina, and vulva or at all four sites simultaneously (Appendix A). Nonetheless, more than one third of HPV 35, HPV 52, and HPV 58 infections occurred among the vulva, vagina, and perianal region, not including cervix (Appendix A).

Therefore, to clarify whether HPV screening at multiple anatomical sites could improve prediction of CIN 2+, we compared the AUC values of the ROC curve for HPV 16/18 detection, which has been recognized as optimally performing in triaging HPV-positive women by an ATHENA study [50,51], observing different site combinations. However, the results demonstrated that the HPV 16/18 co-detected in different combinations of sites did not improve the AUC values compared to cervical HPV 16/18 detection (Table 4).

Furthermore, we analyzed the characteristics of single and multiple HR-HPV infections in CIN 2+ patients (Appendix A). We found that HPV 16 mainly appeared in a single-infection pattern in the vulva, vagina, and cervix, but almost half of HPV 16 in the anus was involved in multiple infections. Regarding other highly prevalent HR-HPVs, such as HPV 52, 58, 53, 33, and 35, multiple infection was their major infection pattern at the four sites, while HPV 56 was present only in multiplex infection. We found 35~50% women with CIN 2+ were HPV 16 negative at the four sites, while HPV 18 was absent in anus of CIN 2+ patients. To further investigate the clinical performance of different HR-HPV combinations screening for predicting CIN 2+, we compared the AUC values of all combinations of eight selected HR-HPVs. Table 4 depicts the top 10 HR-HPV combination tests, with fair AUCs (higher than 0.7) [52]. These combinations clearly gave better results than cervical HPV 16/18, which means higher sensitivity or specificity. Although HPV 33, HPV 35, and HPV 58 testing alone showed poor discrimination, with AUC values around 0.5, their combined detection in a group with HPV 16 could improve the diagnostic performance (details in Appendix A).

## 4. Discussion

HR-HPV genotyping has been proposed for incorporation into primary non-invasive screening for patients at risk of a low level of cervical lesions, such as those with ASC-US or LSIL, based on TCT testing [22,53,54]. Therefore, in the present study, we included those with risks of cervical lesions based on the routine cytology and HPV screening test to demonstrate the HR-HPV infection feature and analyze the association with cervical lesions. Among the 499 participants, 412 individuals had at least one subtype of HR-HPV infection, resulting in an overall infection rate of 82.57%, and 41.7% (208/499) patients showed abnormality (ASC-US or higher levels) in the cytological test. The relatively high HR-HPV infection rate and abnormal cytology rate indicate that our study subjects are those with risks of cervical lesions, most of whom will be commonly advised to undergo colposcopy for further clinical triaging.

From the data of the four detection sites, the main subtypes of HR-HPV were HPV 16, 52, 53, 51, 58, and 56. For each testing site, HPV 16, 52, 53, 51, 58, and 56 occupied the top six HR-HPVs based on occurrence frequency, though they were ranked differently. It has been documented that HPV 16, 18, 31, 33, 35, 52, and 58 are the most commonly detected HR-HPVs in cervical cancer [55], so our prevalence data reinforce the importance of HPV 16, 52, and 58 when it comes to risk management in Chinese women. Furthermore, HPV 16 and 52 were more common subtypes in vaginal, vulvar, and anal sites. Although HPV 18 has been considered an important HR-HPV in cervical cancer, the prevalence of HPV 18 infection was relatively low in our study population. Several studies reported that the most common HR-HPV subtypes were HPV 52, 58, and 16 for Chinese women, in which HPV 18 was not included [56,57,58,59,60,61,62]. In addition to HPV 16, 52, and 58, HPV 53, 51, and 56 were proposed as predominant subtypes of HR-HPVs in Chinese women, because of their relatively high prevalence at all testing sites, among which HPV 53 has been highlighted by other studies on the co-infection status in Chinese women [63]. 

Based on the major subtypes of HR-HPV identified in this study, we analyzed the consistency of seven selected HR-HPV infections (including the aforementioned top six HR-HPVs and HPV 18) between different anatomical sites. To the best of our knowledge, this is the first report on the characteristics of HR-HPV infection at the vulvar and anal sites of Chinese women. In line with the findings in the literature [7], we found that vaginal samples presented moderate to substantial concordance with vulvar and cervical samples, which suggested that the HR-HPV infection features in vagina could, to some extent, reflect the situation at the vulva and cervix. Thus, HR-HPV genotyping might be preferentially performed on vaginal samples when the medical condition is too limited or for patients whose cervix is prone to bleeding. The anal samples presented relatively poor concordance with vulvar, vaginal, and cervical samples, mainly due to the absence of certain HR-HPV subtypes of infection in anus, such as HPV 18 and 35. With regard to the most important HR-HPV, HPV 16, we found that vaginal and cervical infections were identical (κ = 0.829), which is partly consistent with the HPV 16/18 infections reported by Cho et al. (κ: 0.81 to 0.86) [16]. However, unlike the finding regarding HPV 16, the infection of HPV 18 at vagina and cervix was only in a moderate concordance (κ = 0.502) in our study. The low concordances of HPV 18 infection might be due to the relatively low prevalence of HPV 18 infection at anus found in our study. As for other predominant subtypes of HR-HPVs (HPV 51, 52, 53, 56, and 58), the infection concordance between four testing sites was comparable to that of HPV 16.

In cervical squamous carcinoma, co-infection of certain HR-HPVs has been found, which might be due to the genetic similarity of the L1 region of the HPV genome [64,65]. The HPV (52 + 58) was proposed as a dual-infection cluster of HR-HPVs in this study. Since HPV 52 and 58 came from HPV α9 cluster (including HPV 16, 31, 33, 35, 52, and 58), the newly found cluster further supports the phylogenetic similarity of HPV 52 and 58. Apart from HPV (16 + 52), HPV (52 + 58), HPV (53 + 58), and HPV (52 + 53) dual-infection clusters proposed by other studies on Chinese women [66], we proposed new dual infection clusters HPV (16 + 53) and HPV (52 + 56), which were found as the most common co-infections in vagina and cervix, respectively. Additionally, a triple-infection cluster, HPV (52 + 53 + 56), was found at all testing sites in parallel with a CIN 2+ level of pathology. Due to the limitation of the positive sample size, it is difficult to further determine the significance of this triple-infection cluster. However, the reason why HPV 53 and 56 form dual- or triple-infection clusters with HPV 16 or 52 deserves more investigation due to their different genetic backgrounds. Although HPV 16 was present in some multiplex infections, single infection was found to be the dominant mode of infection for HPV 16 at all testing sites and for 18 at cervix, vagina, and anus; the other 13 HR-HPVs showed a higher probability of multiple infections at all testing sites than with single infection. In brief, our results highlighted the site-specific prevalence of different HPV subtypes, which might be caused by the differences in anatomical characteristics in terms of clearance rates of HPV infection or infection process from sources [67]. Future investigations into site-specific HPV subtype distribution and potential mechanisms are still necessary to validate these findings, in order to seek more efficacious interventions to prevent HPV-related cancers.

As mentioned, our study subjects were those at risk of having cervical lesions. Based on colposcopy and pathology studies, we found consistent results, with nearly half (48.7%) of participants with cervical lesions (CIN 1+) and 11.6% with CIN 2+ lesions requiring further treatment, such as cervical conization [68]. From the literature, it is suggested that cervical HPV 16 and HPV 18 are associated with 70% of all cases of invasive cervical cancer, and HPV 16 is the most common high-risk subtype, causing over half of all cervical cancer cases [69,70]. Our results showed that cervical HPV 16/18 detection was still not sufficient for predicting CIN 2+ lesions, with 1/3 CIN 2+ patients not covered. Based on our data, strategies for non-HPV 16/18 HR-HPV detection at different anogenital sites are essential to improve both the CIN 2+ detection rate and efficiency of cervical cancer screening. Herein, we sought a new clinical strategy by considering anogenital sites and HR-HPV subtype simultaneously. However, in our study, there were no significant differences in AUC values of CIN 2+ prediction by HPV 16/18 detection between multiple sites and cervix-only. Although HR-HPV detection at multiple sites was considered to reflect the infection persistence [71], it might not be helpful for reducing false negative and positive rates of CIN 2+ prediction. Nevertheless, we found potential HR-HPV subtypes in combination with HPV 16/18 for improving prediction accuracy. By analyzing the single- and multiple-infection characteristics of the different subtypes in CIN 2+ patients, eight HR-HPVs were selected as candidate indicators. HPV 51, a subtype among the top six high-prevalence subtypes, was ruled out because of its relatively high occurrence rate in patients with ≤CIN 1. In contrast, HPV 33 and 35 were included because of their relatively low occurrence rate in patients with ≤CIN 1 and relatively high occurrence in patients with CIN 2+, although they did not belong to the top five high-prevalence subtypes. As the most commonly detected HR-HPVs in cervical cancer, HPV 16, 18, 33, 35, 52, and 58 were included, whose biological plausibility associated with the etiology of cervical cancer has been extensively documented [72,73]. Our results demonstrated that the co-testing of cervical HPV 33/35/52/53/56/58 with HPV 16/18 had better AUC values compared to the clinically well-recognized cervical HPV 16/18 detection. At present, five types of HPV vaccines have been approved: HPV-2 (16, 18) from three different manufacturers, HPV-4 (6, 11, 16, 18), and HPV-9 (6, 11, 16, 18, 31, 33, 45, 52, 58) [74]. Our results also highlighted the significance of HPV 35/53/56 in cervical lesions, although their association with high-grade lesions is not as clear as that of HPV 16/18/52/58. Further studies are still necessary to evaluate and clarify the clinical effectiveness of these HPV screening combinations in order to provide more evidence of HPV vaccine development for Chinese women.

Additionally, age [75], geographic origin [76], sexual behavior [77], and HPV vaccination [78] might interact with HPV prevalence and thus influence high-grade cervical disease development. HPV infection was not detected in one CIN 2+ patient, which implies that other potential confounding factors might contribute in this patient’s cervical lesion. However, due to the limited CIN 2+ case number (58 cases), we were not able to analyze the possible impact of these confounding factors on the results. Hence, it is still necessary to carry out further work, such as investigating related mechanisms behind the site-specific prevalence of different HPV subtypes, especially in high-risk populations, through multicenter trials.

## 5. Conclusions

Our study provides occurrence characterization of 15 HR-HPVs at four anogenital sites in a group of Chinese women at risk of having cervical lesions. We observed that HPV 16, 52, 53, 51, 58, and 56 were ubiquitous and the top six HR-HPV subtypes infecting the vulva, anus, vagina, and cervix of the subjects. The vaginal samples showed at least substantial agreement with vulvar and cervical samples in terms of the occurrence of the top six subtypes. Additionally, HPV 16 and 18 were preferentially present in single infections, whereas HPV 51, 52, 53, 56, and 58 were mainly found in multiple infections. Moreover, two dual-infection clusters, including HPV (16 + 53) in vagina and HPV (52 + 56) in cervix, and a triple-infection cluster HPV (52 + 53 + 56), presented relatively high prevalence. Our results underscored the importance of several HR-HPVs (including HPV 16, 18, 33, 35, 52, and 58), which have been reported to be most prevalent in cervical cancer or CIN. On the other hand, certain HR-HPVs with less available data, such as HPV 53 and 56, were proposed as additional important subtypes in CIN 2+. Based on these findings, a potential screening strategy was proposed. Compared with HPV 16/18, the other HPV combinations among HPV 33/35/52/53/56/58 along with HPV 16/18 showed higher sensitivity and specificity than cervical HPV 16/18 detection, which should be taken into account in further developing an HPV-based screening strategy. Lastly, more in-depth studies will be needed to determine the association between the proposed HR-HPV infection group and cervical lesions, even cervical cancer. Moreover, there are still some limitations in our results. Due to the limited amount of data, the possible impact of some factors (i.e., age group, geographic origin, sexual behavior, and HPV vaccination, etc.) on the results could not be investigated. Future investigations are still necessary to validate the conclusions drawn from this study.

## Figures and Tables

**Figure 1 cancers-16-02107-f001:**
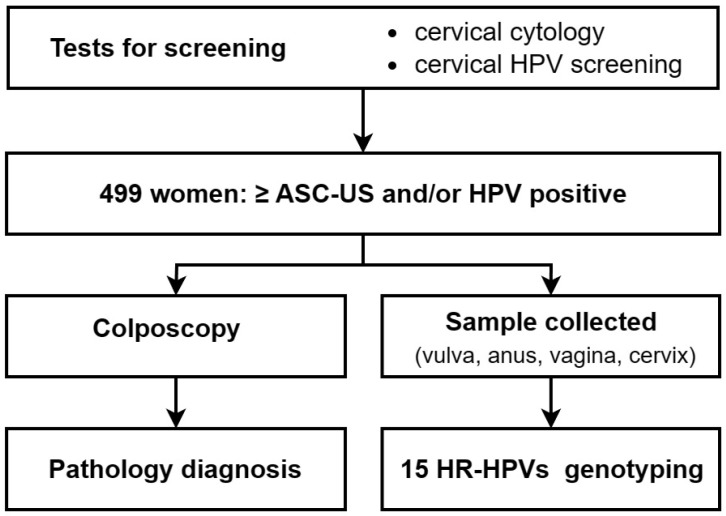
Flowchart of the study.

**Figure 2 cancers-16-02107-f002:**
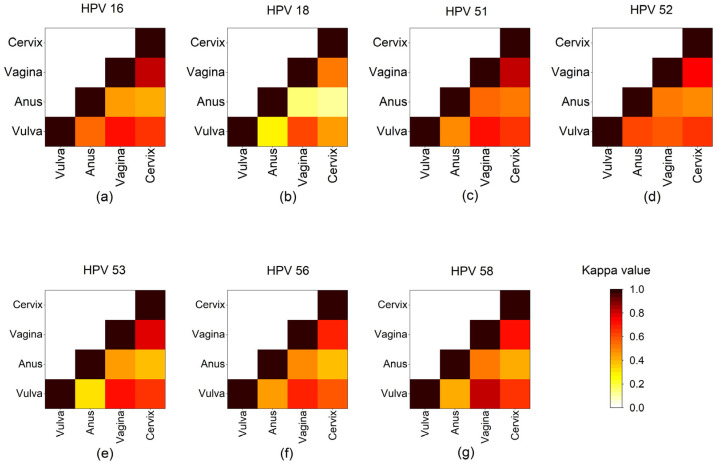
Concordance of seven HR-HPVs detected at different anatomical sites. Heap maps show Cohen’s kappa coefficients (κ) for (**a**) HPV 16, (**b**) HPV 18, (**c**) HPV 51, (**d**) HPV 52, (**e**) HPV 53, (**f**) HPV 56, and (**g**) HPV 58 between two anogenital sites. Each site is represented by a row and a column, and the cells show the concordance between them. Concordance analyses were determined by κ, where κ = 0 means poor, 0 < κ ≤ 0.20 means slight, 0.20 < κ ≤ 0.40 means fair, 0.40 < κ ≤ 0.60 means moderate, 0.6 < κ ≤ 0.80 means substantial, and 0.8 < κ ≤ 1.0 means (almost) perfect agreement [43] The darker the color, the greater the concordance.

**Figure 3 cancers-16-02107-f003:**
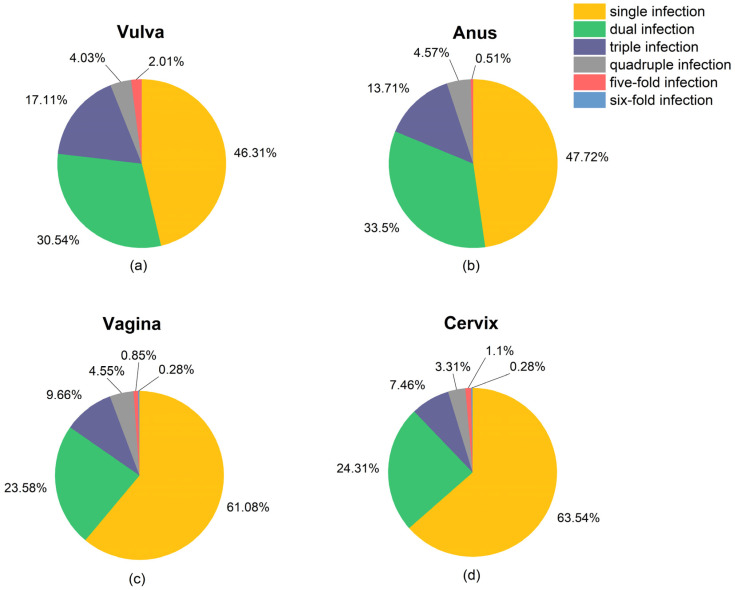
Pie charts present the percentages of single infection and co-infection in (**a**) vulva, (**b**) anus, (**c**) vagina, and (**d**) cervix. Number of patients having infection of the HR-HPVs (≥1 subtype) are 298, 197, 362, and 370 at vulva, anus, vagina, and cervix, respectively.

**Figure 4 cancers-16-02107-f004:**
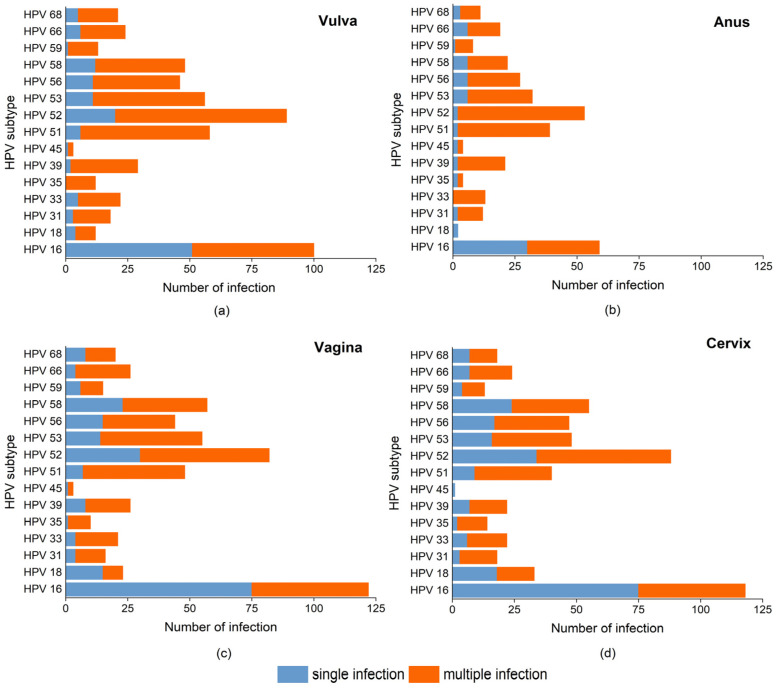
Stacked bar charts show the cases numbers of single and multiple infections of each HR-HPV in: (**a**) vulva, (**b**) anus, (**c**) vagina, (**d**) cervix. The percentage between single and multiple infections of each HR-HPV subtype was compared by Fisher’s exact test, and a *p*-value < 0.05 was considered statistically significant.

**Figure 5 cancers-16-02107-f005:**
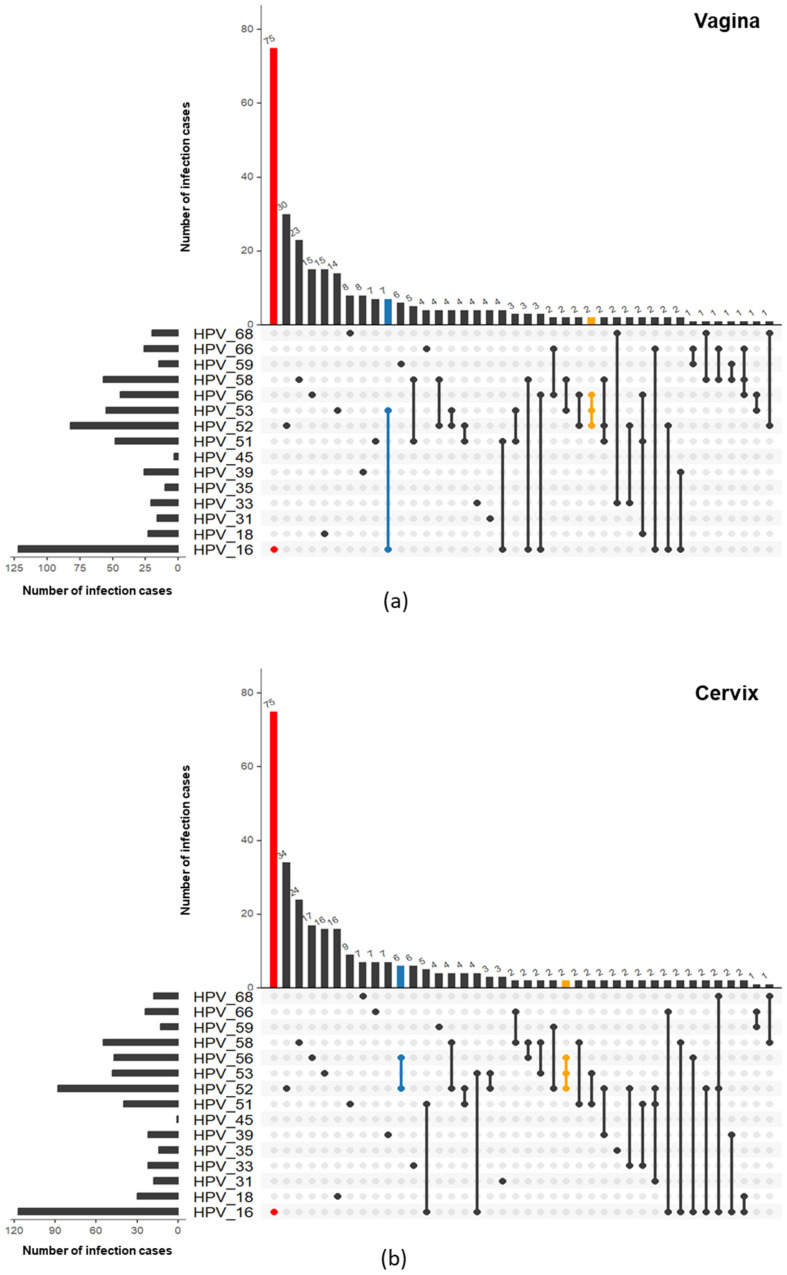
Venn plots show the prevalence of both single- and co-infection patterns in (**a**) vagina and (**b**) cervix based on UpSet results. The total HPV prevalence of each HR-HPV is shown on the left, while the number of each infection pattern is presented on the top, where only the top 40 patterns are shown due to limited space. The most common patterns of single, dual, and triple HR-HPV infections are in red, blue, and yellow, respectively.

**Table 1 cancers-16-02107-t001:** HR-HPV prevalence at four sites.

	HPV Infection No. ^1^	HPV Infection Rate (%)
Number of Infection Sites		
0	90	18.04
1	61	12.22
2	54	10.82
3	139	27.86
4	155	31.06
total	499	100
Anatomical Site		
Vulva	298	59.72
Anus	197	39.48
Vagina	362	72.55
Cervix	370	74.15

^1^ HPV infection No.: number of cases with at least one subtype of HR-HPV infection in different number of infection sites or different anatomical sites.

**Table 2 cancers-16-02107-t002:** Numbers of positive infections of each HR-HPV at four anatomical sites.

HPV Subtype	Anatomical Site				Total
	Vulva	Anus	Vagina	Cervix	
**HPV 16** ^1^	**100 (1)** ^2^	**59 (2)**	**122 (1)**	**118 (1)**	**399**
**HPV 52**	**89 (2)**	**69 (1)**	**82 (2)**	**88 (2)**	**328**
**HPV 53**	**56 (4)**	**32 (4)**	**55 (4)**	**48 (4)**	**191**
**HPV 51**	**58 (3)**	**44 (3)**	**48 (5)**	**40 (6)**	**190**
**HPV 58**	**48 (5)**	**23 (6)**	**57 (3)**	**55 (3)**	**183**
**HPV 56**	**46 (6)**	**27 (5)**	**44 (6)**	**47 (5)**	**164**
HPV 39	29	**23 (6)** ^3^	26	22	100
HPV 66	24	19	26	24	93
HPV 33	22	13	21	22	78
HPV 18	12	2	23	33	70
HPV 68	21	11	20	18	70
HPV 31	18	12	16	18	64
HPV 59	13	8	15	13	49
HPV 35	12	3	10	14	39
HPV 45	3	3	3	1	10

^1^ The top six prevalent HR-HPV subtypes and corresponding cases are listed in bold. ^2^ Numbers in parentheses are ranking numbers for the top six subtypes. ^3^ For anus, the prevalence of HPV 58 is the same as HPV 39, both of which tied for the sixth place.

**Table 3 cancers-16-02107-t003:** The prevalence of 15 HR-HPV in different cervical disease categories.

Rank	Normal (*n* ^1^ = 256)		CIN 1 (*n* ^1^ = 185)		CIN 2+ (*n* ^1^ = 58)	
	Subtype (*n* ^2^)	Count ^3^	Subtype (*n* ^2^)	Count ^3^	Subtype (*n* ^2^)	Count ^3^
1	HPV 52 (71)	164	HPV 16 (51)	134	HPV 16 (38)	116
2	HPV 16 (61)	148	HPV 52 (49)	126	HPV 52 (15)	38
3	HPV 51 (42)	105	HPV 53 (31)	76	HPV 58 (10)	31
4	HPV 58 (42)	94	HPV 56 (32)	73	HPV 56 (10)	22
5	HPV 53 (42)	94	HPV 51 (30)	73	HPV 53 (7)	21
6	HPV 56 (36)	69	HPV 58 (26)	58	HPV 33 (5)	17
7	HPV 39 (25)	62	HPV 66 (15)	41	HPV 35 (6)	14
8	HPV 66 (14)	40	HPV 39 (14)	30	HPV 66 (5)	12
9	HPV 68 (14)	38	HPV 31 (10)	24	HPV 51 (8)	12
10	HPV 33 (15)	38	HPV 68 (9)	23	HPV 18 (4)	11
11	HPV 18 (20)	36	HPV 33 (9)	23	HPV 68 (4)	9
12	HPV 31 (15)	32	HPV 18 (16)	20	HPV 31 (4)	8
13	HPV 59 (10)	25	HPV 59 (6)	20	HPV 39 (6)	8
14	HPV 35 (6)	14	HPV 35 (6)	11	HPV 59 (1)	4
15	HPV 45 (2)	4	HPV 45 (2)	5	HPV 45 (1)	1

CIN = cervical intraepithelial neoplasia; ^1^ the number of patients with one specific cervical disease category; ^2^ number of patients with one specific HR-HPV subtype infection; ^3^ the total number of HPV infections at four anogenital sites in one specific cervical lesion category. The table is sorted by “Count”.

**Table 4 cancers-16-02107-t004:** AUC values of different anogenital sites or HR-HPVs combination tests for predicting CIN 2+.

Tests	AUC Value ^1^	Comment
Cervix (HPV 16/18)	0.723	
Combinations for sites		
Cervix + Anus	0.717	HPV 16/18 detection in different anogenital sites combinations
Cervix + Vulva + Anus	0.704	
Cervix + Vulva	0.699	
Cervix + Vagina	0.693	
Cervix + Vagina + Vulva	0.693	
Cervix + Vagina + Vulva + Anus	0.686	
Cervix + Vagina + Anus	0.681	
Combinations for HR-HPV		
HPV 16/33/58	0.753	different HR-HPV combinations detection in cervix
HPV 16/33/35	0.753	
HPV 16/33/35/58	0.752	
HPV 16/18/33/58	0.750	
HPV 16/18/33/35/58	0.748	
HPV 16/18/33/35	0.747	
HPV 16/33	0.746	
HPV 16/58	0.742	
HPV 16/35/58	0.741	
HPV 16/18/33	0.740	

^1^ AUC = the area under the receiver operating characteristic (ROC) curve. Table 4 is ranked by AUC values.

## Data Availability

The data presented in this study are available on request from the corresponding author.

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
