# Peer review of "High-Risk Genotypes of Human Papillomavirus at Diverse Anogenital Sites among Chinese Women: Infection Features and Potential Correlation with Cervical Intraepithelial Neoplasia"

_cancers, 2024, doi:10.3390/cancers16112107_

Round 1

Reviewer 1 Report

Comments and Suggestions for Authors

The manuscript titled "High-risk genotypes of human papillomavirus at diverse ano-genital sites among Chinese women: infection features and potential correlation with cervical intraepithelial neoplasia", offers a comprehensive analysis of high-risk HPV genotypes across multiple anogenital sites, a topic of high relevance given the established association between HPV and cervical cancers. The findings on the high prevalence and site-related infection consistency of additional subtypes beyond HPV 16/18 are particularly valuable, potentially influencing future screening practices. The use of a large sample size (499 women) and the inclusion of multiple testing sites (vulva, anus, vagina, cervix) provide a robust dataset. The methods are well-described, enhancing the reproducibility of the study. The manuscript addresses a critical gap in the current understanding of HPV infections at diverse anatomical sites and their correlation with cervical lesions, providing crucial data that could lead to more targeted and effective screening strategies.

Aspects for Improvement:

While the statistical methods used (Cohen kappa tests, ROC curve analysis) are appropriate, the manuscript could benefit from a clearer explanation of the statistical choices made, particularly the rationale behind the selection of specific tests and the interpretation of their results.

The manuscript could improve by discussing potential confounding factors that might have influenced the study results, such as the participants' sexual behavior, use of contraceptives, or other health conditions.

The manuscript would benefit from a more detailed limitations section. Discussing the limitations related to the study design, such as potential biases in participant selection or the impact of regional differences within China on the generalizability of the findings, would provide a more balanced view.

While the study provides significant insights, the manuscript could further suggest specific future research directions. For instance, it could propose studies that explore the causal mechanisms behind the site-specific prevalence of different HPV subtypes or evaluate the effectiveness of tailored screening strategies based on the findings.

Some references are outdated or not directly relevant to the study's findings. Updating these references to include more recent studies could strengthen the credibility and relevance of the background and discussion sections.

Overall, the manuscript provides valuable data that contribute to the understanding of HPV's epidemiology and its implications for cervical cancer screening. Addressing these comments could enhance the clarity and impact of the findings.

Comments on the Quality of English Language

Can be improved!

Reviewer 2 Report

Comments and Suggestions for Authors

Estimated Authors,

I've read with interest your article on the high‐risk (HR) genotypes of HPV at diverse anogenital sites among Chinese women. In this study, a total of 499 women were included in the pooled analyses. Authors were able to document HPV 16, 52, 56, and 58 as top five prevalent subtypes in CIN2+, and that HPV 16/18 are mostly associated with HPV 33,35,52,53,56,58. Moreover, will distinctive pattern was identified in the documented sites (Figure 2 and Figure 3). 

The content of the paper is, therefore, both consistent in scope and quality of CANCERS, but some improvements could be envisaged.

First and foremost, Authors should explain how the patients were identified: the parent healthcare center has some specificities in the assisted patient population, either in terms of demography or socio-economic status? This could be of some interest in the interpretation of included results.

Second: the results are of certain interest in terms of preventive interventions, as among the currently available HPV vaccines, the broader coverage is guaranteed by Gardasyl9 encompasses (serotypes 6, 11, 16, 18, 31, 33, 45, 52, and 58). As a consequence, Authors should tentatively reconcile their results with the potential vaccination strategies to be implemented.

Third, Authors should also discuss whether the distribution in documented serotypes is affected by any demographic features (i.e. age groups, geographic origin, socioeconomic status).

After such improvements, I'm confident about the eventual acceptance of this study.

  Comments on the Quality of English Language

Only minor typos scattered across the main text, all of them could be amended by authors themselves by double checking during the resubmission procedures.

Author Response

请参阅附件。

Reviewer 3 Report

Comments and Suggestions for Authors

Zhao et al., in the present manuscript investigated the association of human papillomavirus (HPV) infection at different ano-genital sites with cervical lesions. The authors aimed at identifying if other16/18-HPV subtype(s) may be clinically relevant as additional indicator(s) to improve CIN 2+ screening. About 500 Chines women were employed in this study and the infection of 15 high risk HPVs in vulva, anus, vagina, and cervix was performed to evaluate the association of CIN lesion with high risk HPV subtypes. Zhao et al., found that HPV 51, 53, and 56 had high-frequency at the anogenital sites, in multi-infection patterns, in addition to the well-known cervical cancer-associated HPV 16, 52, which represented the top five prevalent subtypes in patients with CIN 2+. Moreover, it was found that cervical HPV 33/35/52/53/56/58 co-testing with HPV 16/18 may improve CIN 2+ screening performance, thus providing new insights into HR-HPVs screening strategy.

The manuscript is interesting, well written and organized. The findings are clinically relevant.

However, I have some concerns:

-       How does the incidence of lesions at diverse ano-genital sites change with age?

-       Could the authors discuss the benefits (or limitations) of employing HPV vaccination?

-       In the Discussion, besides the need of a larger sample size the authors should also mention the need of multiple centers studies.

-       Do the authors perform any IHC analysis? If available, the manuscript would benefit of this additional information.

Round 2

Reviewer 2 Report

Comments and Suggestions for Authors

The paper has been amended according to my previous comments. Even though there is some misunderstanding regarding my first point from the comments (I did ask Authors to provide some information about the patients usually cared within the parent center), the amendments of the text have made those comments redundant. Therefore, I'm advocating the acceptance of this study.

Reviewer 3 Report

Comments and Suggestions for Authors

The authors have addressed all the concerns.